https://doi.org/10.1038/s42003-020-0895-3　**OPEN**

# The [4Fe-4S] cluster of sulfurtransferase TtuA desulfurizes TtuB during tRNA modification in *Thermus thermophilus*

Minghao Chen [1,7], Masato Ishizaka[2,7], Shun Narai[2,7], Masaki Horitani [3], Naoki Shigi [4], Min Yao[1✉] & Yoshikazu Tanaka [1,5,6✉]

TtuA and TtuB are the sulfurtransferase and sulfur donor proteins, respectively, for biosynthesis of 2-thioribothymidine (s²T) at position 54 of transfer RNA (tRNA), which is responsible for adaptation to high temperature environments in *Thermus thermophilus*. The enzymatic activity of TtuA requires an iron-sulfur (Fe-S) cluster, by which a sulfur atom supplied by TtuB is transferred to the tRNA substrate. Here, we demonstrate that the Fe-S cluster directly receives sulfur from TtuB through its inherent coordination ability. TtuB forms a [4Fe-4S]-TtuB intermediate, but that sulfur is not immediately released from TtuB. Further desulfurization assays and mutation studies demonstrated that the release of sulfur from the thiocarboxylated C-terminus of TtuB is dependent on adenylation of the substrate tRNA, and the essential residue for TtuB desulfurization was identified. Based on these findings, the molecular mechanism of sulfur transfer from TtuB to Fe-S cluster is proposed.

[1] Faculty of Advanced Life Science, Hokkaido University, Kita 8, Nishi 5, Kita-ku, Sapporo, Hokkaido 060-0810, Japan. [2] Graduate School of Life Science, Hokkaido University, Kita 8, Nishi 5, Kita-ku, Sapporo, Hokkaido 060-0810, Japan. [3] Faculty of Agriculture, Department of Applied Biochemistry and Food Science, Saga University, 1 Honjo-machi, Saga 840-8502, Japan. [4] Biotechnology Research Institute for Drug Discovery, National Institute of Advanced Industrial Science and Technology, 2-4-7 Aomi, Koto-ku, Tokyo 135-0064, Japan. [5] Graduate School of Life Sciences, Tohoku University, 2-1-1 Katahira, Aoba-ku, Sendai 980-8577, Japan. [6] Japan Science and Technology Agency, PRESTO, 2-1-1 Katahira, Aoba-ku, Sendai 980-8577, Japan. [7] These authors contributed equally: Minghao Chen, Masato Ishizaka, Shun Narai ✉email: yao@castor.sci.hokudai.ac.jp; yoshikazu.tanaka@tohoku.ac.jp

Sulfur modification is one of the posttranscriptional modifications essential for the correct functioning of transfer RNA (tRNA)[1–3]. It has many important cellular roles, such as codon recognition[4–6], thermostability[7,8], and ultraviolet irradiation sensing[9]. The biosynthesitic pathway for sulfur modification of tRNA has been the subject of investigation over several decades, and recently a number of sulfur transferases have been identified that require an inorganic cofactor, an iron-sulfur (Fe-S) cluster, for their enzymatic activity. Examples include: TtuA, which catalyzes the formation of 2-thioribothymidin (s²T) at position 54 (refs. [10,11]); TtcA, which catalyzes the formation of 2-thiocytidine (s²C) at position 32 (ref. [12]); and ThiI, which catalyzes the formation of 4-thiouridine (s⁴U) at position 8 (ref. [13]). This has stimulated new studies investigating the mechanism of the Fe-S cluster-dependent sulfur transfer reaction.

TtuA is the best-characterized enzyme among the Fe-S-dependent sulfur transferases[10,11,14–16] and is known to contain a [4Fe-4S] type cluster in the catalytic site (Fig. 1). Only three cysteine residues chelate the cluster, leaving one Fe atom (hereafter referred to as the unique Fe site) remaining free from coordination by TtuA. In a recent study using X-ray crystallography, an extra electron density was discovered bound to the unique Fe site of the [4Fe-4S] cluster of *Pyrococcus horikoshii* TtuA. The atom was not identified but, because the electron density was more consistent with a hydrosulfide rather than a hydroxide ion, Arragain et al. proposed that the unique Fe site of TtuA is responsible for capturing the sulfide ion and transferring it to the substrate tRNA (ref. [11]). In their proposed mechanism, sulfur is first trapped on the unique Fe site of the [4Fe-4S] cluster (hereafter this state is referred to as the [4Fe-5S]-intermediate), from which it is transferred to tRNA.

TtuB is a ubiquitin-like protein with a molecular mass of ~7 kDa. It functions as an intrinsic proteinous sulfur donor in the TtuA reaction[14], supplying a sulfur atom as a thiocarboxylate moiety at its C-terminus, TtuB-COSH (refs. [15,17,18]; Fig. 1). Similar usage of a thiocarboxylate moiety to protect and transport a sulfur atom is known also for the biosynthetic pathways of both thiamin[19] and molybdenum cofactor[20]. However, for both pathways, the sulfur atom is transferred directly from the donor protein to the substrate, so an Fe-S cluster is not required. Therefore, the molecular basis for the transfer of sulfur from TtuB to tRNA via the Fe-S cluster is still unclear.

Sulfide can be used as a substitute sulfur donor for s²T formation, particularly in some organisms in which the *ttub* gene is absent, such as *Thermotoga maritima*. These organisms are reported to achieve direct use of sulfide for s²T formation by maintaining a high cytoplasmic sulfide concentration[21,22]. However, TtuB is an irreplaceable sulfur donor for in vivo s²T biosynthesis in organisms possessing the ttub gene. For example, ablation of the *ttub* gene completely impairs the formation of s²T in *Thermus thermophilus*[8]. Organisms with the *ttub* gene are unable to maintain a high cytosolic sulfide concentration due to its toxicity[23]. Therefore, TtuB is probably used for storing, transferring, and providing sulfur, conserving it as a resource and protecting against sulfide toxicity. A set of eukaryotic proteins (homologous to TtuA and TtuB), Ncs6/Ncs2, and Urm1, is responsible for the formation of s²U at position 34 of cytosolic tRNA in eukaryotes[17,18,24–27], suggesting that TtuB-based sulfur transfer is widely conserved. Therefore, investigation of the mechanism of sulfur transfer from TtuB to the Fe-S cluster is essential for a general understanding s²T biosynthesis across a wide range of organisms.

In the present study, we investigate the molecular mechanism of sulfur transfer from TtuB to the Fe-S cluster. Electron paramagnetic resonance (EPR) spectroscopic studies demonstrate that the Fe-S cluster binds directly to TtuB, and desulfurization assays reveal that the sulfur is not released from TtuB immediately after binding. The state in which the Fe-S cluster binds with TtuB was visualized by X-ray crystal structure analysis, which provided a better understanding of the binding properties of the unique Fe site. Desulfurization assays indicated that the release of sulfur from TtuB requires adenylation of the substrate tRNA, and mutational studies identified the critical residue for the TtuB desulfurization reaction. Based on the results, a comprehensive model is proposed for the sulfur transfer reaction achieved by TtuA and TtuB.

## Results

**Crystal structure of TtuA–TtuB–[Fe-S]–ATP complex.** The TtuA possessing the [4Fe-4S] cluster (holo-TtuA) in complex with TtuB was crystallized, in order to observe the coordination of Fe-S in the presence of TtuB. The crystal structure of holo-TtuA in complex with TtuB-COOH and adenosine triphosphate (ATP) was successfully determined at 2.2 Å resolution (Fig. 2a). The manner of interactions between TtuA and Fe-S (i.e., TtuA coordinately bound to Fe-S via three cysteine residues) was identical to that for holo-TtuA reported previously[10]. However, a new interaction directly coordinated between the unique Fe site of the [4Fe-4S] cluster and the C-terminus of TtuB was observed (Fig. 2b). One of the three TtuA–TtuB heterodimers in one asymmetric unit showed the electron density of the Fe-S cluster connected to the C-terminus of TtuB (Supplementary Fig. 1), indicating coordinate binding of TtuB to the Fe-S cluster. Distinct from other Fe atoms belonging to the Fe-S cluster, which are tetrahedrally coordinated, the unique Fe site is octahedrally coordinated to the C-terminal of TtuB (Fig. 2c). Two oxygen atoms from TtuB and two sulfur atoms from the Fe-S cluster form the equatorial plane of the octahedron. The distance between each of the two oxygen atoms and the unique Fe site was 2.4 and 2.5 Å, respectively (Fig. 2c). The substrate binding manner of [4Fe-4S] cluster in TtuA is similar to that of aconitase[28] and radical-S-adenosyl-L-methionine enzymes[29], which use the unique Fe site to place a water molecule necessary for the reaction at an appropriate position. Along the axis of the octahedron, a water molecule was detected ~3.7 Å from the unique Fe site (Fig. 2c). On this axis, Lys137 of TtuA lies across the water molecule and forms a hydrogen bond with it. Based on the similarity of the chemical structure of the carboxyl (-COOH) and thiocarboxyl (-COSH) groups, the structure of TtuA–TtuB (COOH) is considered to mimic the state of TtuB-COSH coordinated by the unique Fe site after forming the TtuA–TtuB complex.

**Identification of [4Fe-4S]–TtuB intermediate.** The coordination states of the Fe-S cluster in the absence and presence of TtuB-COSH (Fig. 3a) were measured by EPR spectroscopy at 12 K,

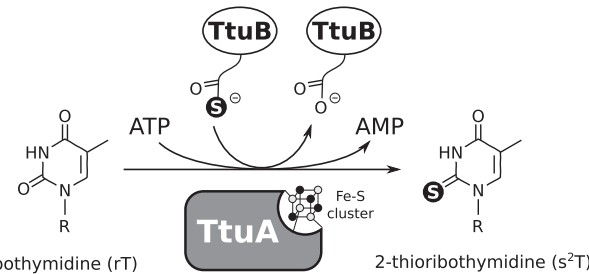

**Fig. 1 Scheme of 2-thioribothymidine biosynthesis.** TtuA transfers sulfur from sulfur donor TtuB to substrate tRNA by consuming ATP. The enzymatic activity of TtuA is dependent on a [4Fe-4S] cluster.

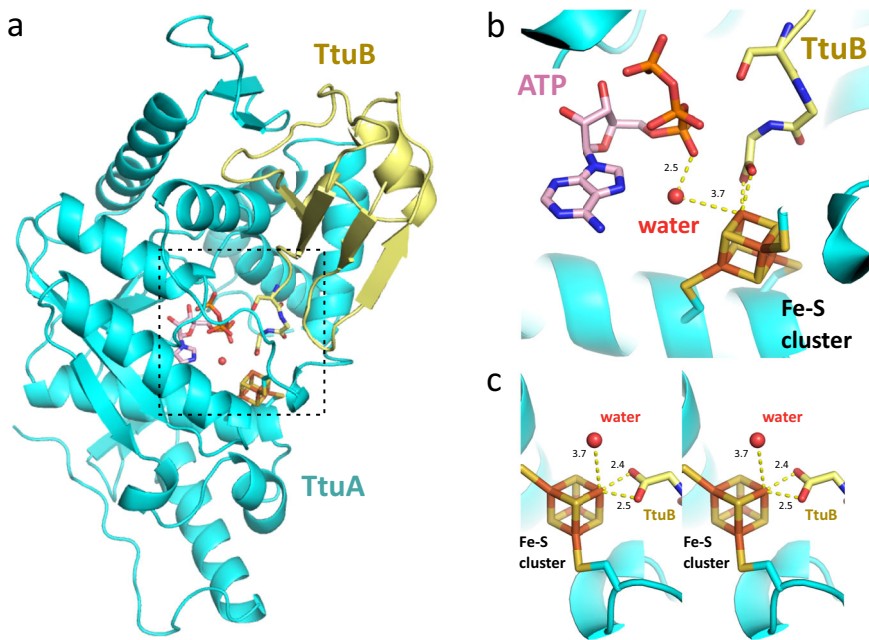

**Fig. 2 Crystal structure of *Tth*TtuA in complex with *Tth*TtuB, Fe-S cluster, and ATP. a** Overall structure of the quaternary complex. TtuA and TtuB are colored as cyan and light yellow, respectively. The catalytic center is indicated by dotted frame. **b** Close-up view of the catalytic center. The Fe-S cluster is represented as a stick model, and colored orange (Fe) and yellow (S). The ATP is represented as stick model as well, and the water molecule is represented as a red ball. **c** Wall-eye stereo view of the Fe-S cluster for showing the octahedral coordination around the unique Fe site. The distances of the coordination bonds around the unique Fe site and the water molecule are indicated in angstrom units.

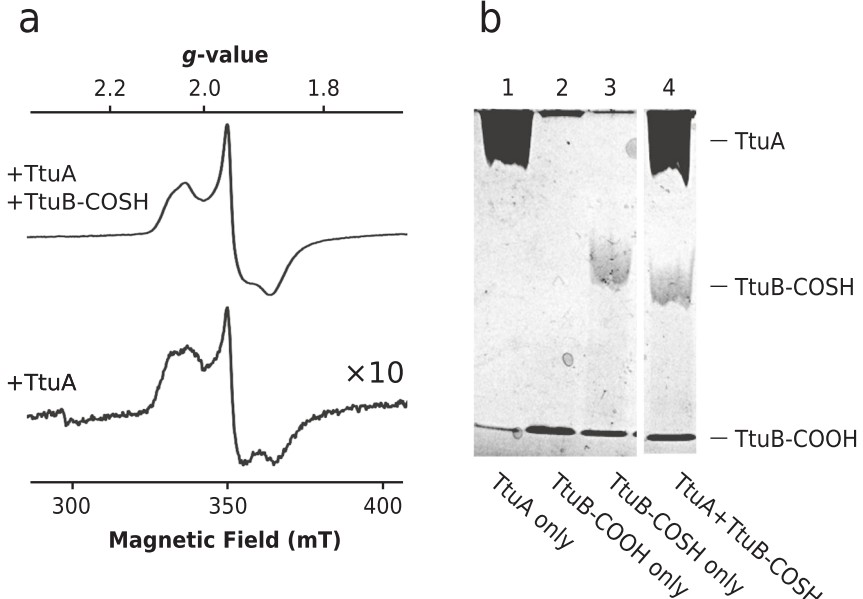

**Fig. 3 Identification of [4Fe-4S]–TtuB intermediate. a** Comparison between EPR spectra of DT-reduced TtuA in the presence (upper) and absence (lower) of TtuB-COSH. TtuA was mixed with TtuB-COSH in a molar ratio of 1:3 (0.6 mM:2.0 mM). The scale of the EPR spectrum of TtuA in the absence of TtuB-COSH is ten times larger than that in the presence of TtuB-COSH, as indicated. The scales of magnetic field and *g*-value are shown and intensities of spectra are adjusted for comparison. **b** TtuB desulfurization assay using APM PAGE. A total of 600 pmol of TtuA (lane 1), TtuB-COOH (lane 2), and TtuB-COSH (lane 3) were loaded. A mixture of TtuB-COSH and TtuA (600 pmol:600 pmol) was loaded (lane 4) after incubation for 30 min at 333 K. The positions of TtuA, TtuB-COOH, and TtuB-COSH are indicated. The complete image of original gel is available in Supplementary Fig. 6a.

which is a typical temperature for observing EPR signals. Since the EPR spectra are too complex to analyze as they stand, their temperature dependence was evaluated by taking measurements at 7 K and 29 K. A relatively simple spectrum was observed at 29 K (Supplementary Fig. 2a), which was interpreted as a mixture of two axial [4Fe-4S] EPR spectra (*g* = [2.074/2.035, 1.943, 1.900];

Supplementary Fig. 2b). These two components were designated conformer 1 and conformer 2, respectively. Similar spectra reported in previous studies of the Fe-S cluster concluded that the spectra are derived from a mixture of two different structures of the [4Fe-4S] cluster[30], or from an identical form of the [4Fe-4S] cluster but in two different reduction states[31]. Accordingly, the

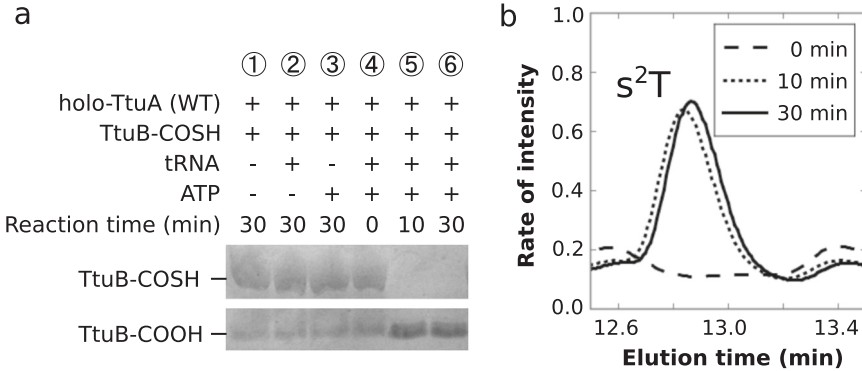

**Fig. 4 Tracing the sulfur atom. a** Results of APM PAGE TtuB desulfurization assay. TtuB-COSH (450 pmol) was incubated with holo-TtuA (75 pmol), and/ or substrate tRNA (600 pmol) and/or ATP (2.5 mM) at 333 K for 30 min. The combinations of each component of each reaction are indicated. The positions of TtuB-COOH and TtuB-COSH are indicated. The complete image of original gel is available in Supplementary Fig. 6b. **b** Results of HPLC TtuA activity assay. The peak of the product $s^2T$ after incubating the reaction mixture for 10 min or longer is shown. The spectra are normalized with the peak of pseudouridine ($\Psi$) as a reference.

spectra obtained for TtuA also could arise from these effects. However, it is deduced that the spectrum at 7 K contains a component other than conformer 1 and conformer 2, since the spectrum at 29 K is not identical to that at 7 K. The characteristic spectrum of this component was estimated by subtracting the spectrum at 29 K from that at 7 K (Supplementary Fig. 2c), with the scale of the two conformer spectra normalized using $g_z$ so as to cancel out conformer 1 and 2 completely. This difference spectrum (hereafter, conformer 3) does not coincide with any typical spectrum for the Fe-S cluster reported previously, so it is presumably composed of the spectra of several unknown conformers. In contrast, the spectrum of TtuA in the absence of TtuB-COSH is the typical rhombic signal of the [4Fe-4S] cluster ($g = [2.055, 1.942, 1.870]$; Fig. 3a, Supplementary Fig. 2d). The difference of the EPR signal between absence and presence of TtuB-COSH suggests that the coordination state of the Fe-S cluster alters on an addition of TtuB-COSH.

The result of EPR spectroscopy (i.e., the EPR spectrum of the TtuA–TtuB(COOH) complex used for X-ray crystallography) supports well the idea that the structure of TtuA–TtuB(COOH) mimics the state of TtuB-COSH coordinated by the unique Fe site. The temperature dependency (Supplementary Fig. 2e) was similar to that of TtuA in the presence of TtuB(COSH). Conformers 1 and 2 were identified by deconvolution of EPR simulation analysis as observed for TtuA in the presence of TtuB-COSH (Supplementary Fig. 2f). This finding suggests that both co-expressed TtuA–TtuB(COOH) and TtuA–TtuB(COSH) have their Fe-S cluster coordinated in the same manner.

It is noteworthy that the formation of a coordination bond stabilizes both TtuB and the Fe-S cluster. The two C-terminal residues of TtuB become disordered in the absence of the Fe-S cluster[10], but their electron density was clearly observed in the present structure (Supplementary Fig. 1c), which suggests that the C-terminus of TtuB is stabilized by the formation of a coordination bond with the Fe-S cluster. Importantly, the intensity of the EPR signal of the Fe-S cluster of TtuA dramatically increased approximately tenfold in the presence of TtuB-COSH (Fig. 3a). This would be caused by alteration of the redox potential of the Fe-S cluster because the cluster became more likely to be reduced by dithionite (DT) and the quantity of reduced [4Fe-4S] cluster was increased when TtuB-COSH was bound. Equally, the Fe-S cluster is stabilized by TtuB. These observations indicate that the conformation of both the TtuB-C-terminus and the Fe-S cluster was stabilized by forming the coordination bond between them.

Subsequently, in order to determine whether the sulfur is removed from TtuB-COSH to form the [4Fe-5S] intermediate or remains on the C-terminus of TtuB, the state of TtuB-COSH was evaluated after binding to TtuA. Polyacrylamide gel electrophoresis (PAGE) containing (N-acryloylamino)phenylmercuric chloride (APM) was used to monitor the sulfur content of TtuB before and after the incubation with TtuA. Because APM reduces the mobility of TtuB-COSH in the gel due to the interaction with the thiol group, TtuB-COSH can be separated from carboxyl TtuB (TtuB-COOH) in the APM gel (Fig. 3b, lanes 2 and 3). The results show that TtuA by itself does not induce desulfurization of TtuB (Fig. 3b, lane 4). Taken together with the EPR spectroscopy and crystal structure described above, this suggests that TtuB is directly coordinated to the [4Fe-4S] cluster via the unique Fe site, but the sulfur is not mobilized. Therefore, it can be concluded that upon TtuB-COSH binding to TtuA, an intermediate is formed in which the [4Fe-4S] cluster is coordinated with the C-terminus of TtuB. This state is referred to as the [4Fe-4S]–TtuB intermediate.

**Desulfurization of TtuB requires tRNA adenylation.** To further understand the reaction mechanism of TtuA, the conditions required for the release of sulfur from TtuB were investigated. Using the APM contained gel, the sulfurization of TtuB was evaluated after the incubation with TtuA (Fig. 4a, lane 1) and additional substrate tRNA (lane 2), ATP (lane 3), or both (lanes 4–6). TtuB desulfurization occurred only when tRNA, TtuA, and ATP were added together and incubated for 10 min or longer (lanes 5 and 6). Accordingly, the final product, $s^2T$, was detected in conditions in which sulfur was mobilized from TtuB (Fig. 4b), confirming that the mobilized sulfur was transferred to the tRNA. These observations demonstrate that the binding of TtuB-COSH to TtuA is not sufficient to induce desulfurization, but rather that all of the components, including TtuA, TtuB-COSH, substrate tRNA, and ATP are necessary for desulfurization.

Next, the charged and conserved residues surrounding the active site of TtuA were substituted with alanine, and the enzymatic activity of these mutants was evaluated (Fig. 5a). The activities of S55A, D59A, K137A, and D161A were dramatically decreased (Fig. 5b). Subsequently, in order to identify the essential residue for the mobilization of sulfur from TtuB-COSH, the activity of each of the four inactive alanine-substituted mutants was evaluated in the presence of 10 mM $Na_2S$ instead of TtuB-COSH. Interestingly, K137A is the only variant that is able to use $Na_2S$ as an alternative source of sulfur to complete the

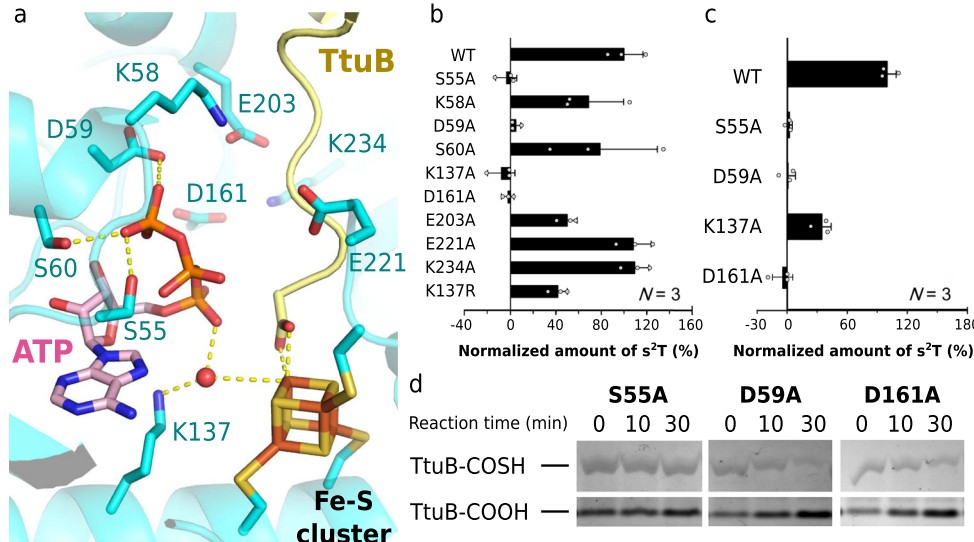

**Fig. 5 Mutation study of TtuA. a** Close-up view of the catalytic center of TtuA. The residues substituted with alanine are indicated. A proton lies between D59 and the γ-phospho group of ATP is not shown. ATP, TtuB, and the Fe-S cluster are represented as described in Fig. 1b. **b** Comparison of wild-type (WT) and mutant TtuA enzymatic activities using TtuB-COSH as a sulfur donor. The amount of $s^2T$ from each mutant after a 30-min reaction is compared with WT. Measurements of each mutants were using same sample and individually measured in triplicate. All the individual data points are shown with the means ± SD. Raw data of HPLC chromatograms are available in Supplementary Data 1. **c** Comparison of WT and mutant TtuA enzymatic activities using 10 mM $Na_2S$ as a sulfur donor. Measurements of each mutants were using same sample and individually measured in triplicate. All the individual data points are shown with the means ± SD. Raw data of HPLC chromatograms are available in Supplementary Data 2. **d** TtuB desulfurization assay with TtuA mutants in the presence of tRNA and ATP. Incubation (reaction) time is indicated above the lanes. Other reaction conditions are the same as those of Fig. 4a, lanes 4–6. The complete image of original gel is available in Supplementary Fig. 6c.

overall reaction among above four alanine-substituted mutants (Fig. 5c). These results indicate that the side chain of the Lys137 is essential for the mobilization of sulfide from TtuB-COSH. That is, although the sulfur-mobilizing activity of the K137A mutant was diminished due to the absence of the side chain, $s^2T$ formation activity was restored because the mobilization of sulfur from TtuB-COSH is not necessary in the presence of $Na_2S$. It is noted that K137R showed enzymatic activity using TtuB-COSH as a sulfur donor (K137R of Fig. 5b). These results suggest that a positively charged side chain at position 137 is essential for the sulfur-mobilizing activity of TtuA. In contrast, mutants S55A, D59A, and D161A did not show enzymatic activity even in the presence of $Na_2S$, excluding them from possible residues involved in the mobilization of sulfur from TtuB-COSH. S55 and D59 are located in the PP-loop motif, widely known as a conserved protein motif for ATP hydrolysis[32,33]. Also, in vivo mutational analyses have confirmed that D161 is a critical residue for ATP binding/hydrolysis[16,34]. Lack of these residues should lead to the loss of tRNA adenylation activity of TtuA, and eventually abolish the $s^2T$ formation reaction.

The ability of S55A, D59A, and D161A to mobilize sulfur from TtuB-COSH was further evaluated and found to be lower for all three mutants (Fig. 5d). In the crystal structure, Ser55, Asp59, and Asp161 are located out of range of direct interaction with the carboxyl terminus of TtuB (farther than 7 Å), suggesting that diminished desulfurization of these mutants was not due to an inability for direct interaction with the C-terminus of TtuB. These residues are known to be involved in ATP hydrolysis, as mentioned above[33]. It is therefore plausible that the abolition of ATP hydrolytic activity (owing to substitution for Ser55, Asp59, and Asp161) caused the dramatic observed decrease of desulfurization.

## Discussion
### Mechanism of sulfur transfer from TtuB to tRNA via Fe-S cluster. The biosynthesis of $s^2T$ is composed of two sequential

reactions: adenylation (Fig. 6e, f) and thiolation (Fig. 6f, g). Adenylation of the substrate is commonly used by a large variety of enzymes. In general, the PP-loop domain is responsible for the adenylation and the Fe-S cluster is not required. Therefore, adenylation of the substrate tRNA by TtuA should be similar to that of other enzymes, in which the Fe-S cluster does not contribute. For the later thiolation step, there are two probable pathways: first is that sulfur is at first trapped on the Fe-S cluster and then transferred to tRNA (hereafter referred to as the indirect pathway); and second is that sulfur is directly transferred from the thiocarboxylated C-terminus of TtuB to tRNA (hereafter referred to as the direct pathway). The indirect pathway (Fig. 6a–c) is considered to be more reasonable because of the following points: first, the sulfur donor protein TtuB is absent in some organisms, including *T. maritima* and *P. horikoshii*; second, even TtuA that has a native-partner TtuB (e.g., *T. thermophilus* TtuA) can transfer a sulfide ion to tRNA; and third, an extra electron density (to which a sulfide ion fits well) was found on the unique Fe site of the [4Fe-4S] cluster of *P. horikoshii* TtuA (ref.[11]).

In the indirect pathway, the intermediate formation of [4Fe-5S] (Fig. 6c) is considered to be essential for thiolation, but the mechanism by which the intermediate is generated from TtuB-COSH on the [4Fe-4S] cluster was unclear. The present study combined X-ray crystallographic, EPR spectroscopic, and biochemical techniques to demonstrate that TtuA performs this reaction by capturing the C-terminus of TtuB directly on the unique Fe site of the [4Fe-4S] cluster (Figs. 2 and 3). The presence of the [4Fe-4S]–TtuB intermediate was identified by EPR spectroscopy and a desulfurization assay, which also revealed that the desulfurization occurs on the Fe-S cluster. This result also excluded the possibility that the sulfur is released from TtuB to be a diffused sulfide ion, which eventually attaches to the unique Fe site. A correlation was identified between TtuB desulfurization and tRNA adenylation. After formation of the [4Fe-4S]-TtuB

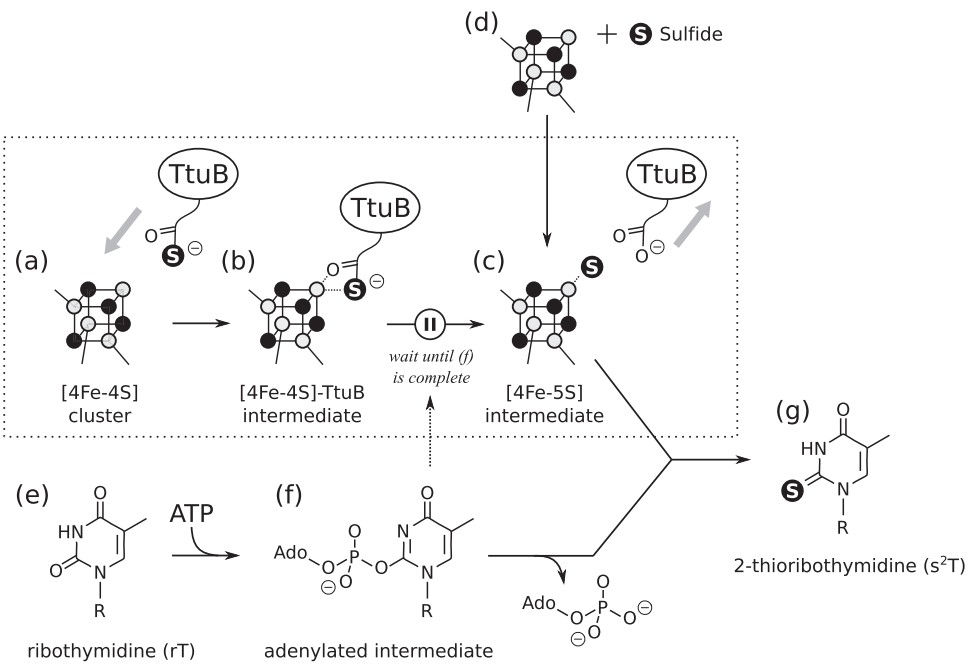

**Fig. 6 Assumed indirect s²T biosynthesis mechanism.** TtuB-COSH accesses the catalytic center of TtuA **a**, and its C-terminus attaches to the Fe-S cluster and forms a [4Fe-4S]–TtuB intermediate **b**. When adenylation is complete (dotted arrow), the sulfur atom is cleaved from TtuB and forms a [4Fe-5S] intermediate **c**. The [4Fe-5S] intermediate also forms in the presence of sulfide as an alternative pathway **d**, which may occur if the organism does not possess TtuB. The substrate tRNA base **e** is activated by ATP to form an adenylated intermediate **f**, which is required for the cleavage reaction of sulfur from TtuB by an as yet unknown mechanism. Ultimately, the sulfur captured by the Fe-S cluster nucleophilically attacks the adenyl group on the ribothymidine (rT), resulting in the formation of the s²T product. The reaction delineated in the present study is indicated by a dotted frame.

intermediate, sulfur is not released from TtuB immediately (Fig. 3b). The reaction probably does not proceed spontaneously in this intermediate state, resuming after adenylation is completed (Fig. 6b, c). Under conditions without ATP or tRNA (Fig. 4a), or using adenylation-deficient TtuA mutants (S55A, D59A, and D161A), desulfurization of TtuB was abolished (Fig. 5d). These observations indicate that desulfurization and adenylation do not occur independently, and that the reaction sequence is precisely controlled. It is probable that desulfurization does not occur until all components are present in the catalytic pocket. Thus TtuB-COSH is used efficiently and there is no sulfide toxicity from inappropriately released sulfur.

**Deduced TtuB desulfurization scheme.** This work focused on identifying the critical residue(s) of TtuA needed for TtuB desulfurization. Firstly, determining the crystal structure enabled elucidation of the mechanism of TtuB desulfurization. Alanine substitution of the charged residues surrounding the catalytic center identified four residues essential for the catalytic activity of TtuA: Ser55, Asp59, Lys137, and Asp161 (Fig. 5b). Since only Lys137 among the four restored the enzymatic activity in the presence of Na₂S instead of TtuB-COSH (Fig. 5c), it was deduced that the side chain of Lys137 is essential for mobilizing sulfur from TtuB. K137R also showed enzymatic activity (Fig. 5b), so an amino or guanidino group at the position of residue 137 would be necessary for mobilizing sulfur from TtuB-COSH. Indeed, Lys137 of TtuA is substituted for arginine in other organisms, including *T. maritima*, emphasizing that a basic residue at this site is important for s²T formation.

Crystal structure showed that a water molecule located 3.7 Å away from the unique Fe site is coordinated with the α-phosphate of ATP (Fig. 5a). It is expected that the α-phosphate of ATP would shift after adenylation of the substrate base. This movement of ATP α-phosphate may push the water molecule close to

the unique Fe site, potentially initiating the desulfurization reaction. Deprotonation of this coordinated water molecule would be the first step for TtuB desulfurization and this could be performed by the basic amino group of Lys137, which can be substituted by arginine (Fig. 5b). Notably, the activity of Lys137 was not fully restored even though Na₂S was supplied, suggesting that Lys137 might be involved also in other processes, such as adenylation.

Based on the findings in the present study, we propose the following reaction scheme for TtuB desulfurization (Fig. 6). It should be noted that this mechanism assumes the indirect pathway for the sulfur transfer from TtuB-COSH to tRNA because it is the most plausible pathway considering the previous experimental data. The C-terminus of TtuB-COSH directly binds to the unique Fe site of the Fe-S cluster (Fig. 6a), which forms a [4Fe-4S]-TtuB intermediate (Fig. 6b). Desulfurization does not occur spontaneously, and this state is maintained until adenylation of tRNA. Adenylation of tRNA then initiates desulfurization, pushing a water molecule closer to the unique Fe site, due to the movement of the AMP generated. The amino group of Lys137, located close to the active site, deprotonates the water molecule and the hydroxide ion generated nucleophilically attacks a carbon of the thiocarboxyl group of TtuB (Supplementary Fig. 3). A nucleophilic elimination reaction on the thiocarboxyl moiety cleaves the sulfur from TtuB, which finally generates TtuB-COOH and the [4Fe-5S] intermediate (Fig. 6c).

It is worth mentioning an alternative mechanism (the direct mechanism; Supplementary Fig. 4) whereby sulfur is considered to be transferred from a [4Fe-4S]–TtuB intermediate directly to tRNA. The free Fe site of the [4Fe-4S] cluster would be used to coordinate the oxygen atom of the thiocarboxylate group and present the sulfur to the substrate tRNA. The carbonyl carbon will be more positively charged due to coordination to the [4Fe-4S] cluster by the inductive effect, which may further enhance the

release of sulfur to attack the adenylated tRNA. The subsequent elimination of sulfur from the carbonyl group would be carried out by a nucleophilic attack of the water coordinated between free Fe and Lys137. Based on the data obtained in the present study, the possibility that the direct mechanism occurs cannot be excluded. Indeed, it provides a more reasonable explanation than the indirect mechanism because the TtuB desulfurization reaction halts until the tRNA adenylation reaction takes place (Supplementary Fig. 4b′–f). However, because the native-partner TtuB (crucial for the direct mechanism) does not exist in all the organisms in which this reaction occurs, it is probably not the primary mechanism.

In conclusion, the present study provides insights into the involvement of the Fe-S cluster on desulfurization of TtuB-COSH. Structural analysis demonstrated that the unique Fe site of the [4Fe-4S] cluster binds directly to the C-terminal thiocarboxylate group of TtuB to form the [4Fe-4S]–TtuB intermediate, indicating that desulfurization occurs on the unique Fe site. A TtuB desulfurization assay revealed that desulfurization does not occur upon binding to TtuA, but on coupling with the sulfur transfer to tRNA. Mutation analysis identified the residue essential for TtuB desulfurization as Lys137. The reaction pathway is summarized in Fig. 6 and is assumed to be indirect (previous research has found that either hydrosulfide or hydroxide is coordinated on the unique Fe site, which agrees well with the indirect pathway[11]). Substrate tRNA is adenylated via an ATP pyrophosphatase used in TtuA (refs. [32,33]). The most plausible reaction mechanism for TtuA to remove sulfur from TtuB-COSH is therefore proposed to be as follows: (1) TtuA captures the C-terminal of TtuB with an Fe-S cluster; (2) waits until adenylation is complete; and (3) releases sulfur from TtuB by nucleophilic attack with a hydroxide ion, benefiting from a charge on the side chain at Lys137. (4) Finally, the sulfide bound to the unique Fe site attacks the adenylated tRNA and substitutes the adenyl group.

## Methods

**Cloning, expression, and purification of the TtuA–TtuB complex.** DNA fragments encoding TtuA and TtuB from *T. thermophilus* (HB27 strain) were cloned into a pETDuet-1 vector at the NdeI/BglII and NcoI/BamHI sites, respectively (Novagen, Madison, Wisconsin, USA). A His6-tag with an additional four residues was attached to the N-terminus of TtuB. TtuA and TtuB were co-expressed by *Escherichia coli* strain B834 (DE3) harboring this modified expression vector following growth in Lysogeny broth medium containing 100 mg/l ampicillin at 310 K, with agitation at 150 rpm. When absorption at 600 nm reached 0.6, isopropyl β-D-1-thiogalactopyranoside was added to a final concentration of 0.1 mM. The culture was then incubated at 298 K for an additional 20 hours and *E. coli* cells were collected by centrifugation at 4500 × g for 30 min. The cells were stored at 193 K until lysis.

The TtuA–TtuB complex was purified in anaerobic conditions with 5% hydrogen and 95% nitrogen gas (Vinyl Anaerobic Chamber, COY, Grass Lake, Michigan, USA), as TtuA has a strong propensity to precipitate in aerobic conditions[10]. The collected cells were sonicated on ice in sonication buffer consisting of 50 mM HEPES-KOH (pH 7.6), 200 mM ammonium sulfate, 50 mM ammonium acetate, 5 mM magnesium chloride, 10% (v/v) glycerol, 7 mM 2-mercaptoethanol, and 0.1% Triton X-100. Following heat treatment at 343 K for 20 min, the precipitate was removed by centrifugation at 7000 × g for 60 min. The supernatant was then loaded onto a His-Trap HP column (GE Healthcare, Chicago, Illinois, USA) pre-equilibrated with the purification buffer (the sonication buffer without Triton X-100). Nonspecifically adsorbed proteins were removed using wash buffer (purification buffer containing 50 mM imidazole). Subsequently, the target protein was eluted with a 50–500 mM gradient of imidazole in purification buffer, and further purified with a HiLoad 16/60 Superdex 200 column (GE Healthcare) pre-equilibrated with sonication buffer. Purity of the sample was confirmed by 15% (v/v) sodium dodecyl sulfate (SDS)–PAGE and luminescent images were analyzed by a LuminoImager (LAS-3000; Fuji Film Inc.).

**Reconstruction of the Fe-S cluster in the TtuA–TtuB complex.** Dithiothreitol was added to the purified TtuA–TtuB complex to a final concentration of 5 mM. The concentration of TtuA–TtuB complex was decided by Nanodrop DU 1000U (Thermo Fisher, Waltham, Massachusetts, USA). After a 10 min incubation, a 20-fold molar excess of ferric chloride (III) was added, followed by a further 10 min incubation. Subsequently, a 20-fold molar excess of sodium sulfide was added,

followed by incubation for 3 hours at 298 K. The iron sulfide precipitate was removed by filtration, and excess iron and sulfide ions were then removed on a Sephadex NAP-25 desalting column (GE Healthcare, Chicago, Illinois, USA). All of the treatments were conducted in anaerobic conditions.

**EPR spectroscopy.** Purified holo-TtuA was concentrated and mixed with TtuB-COSH (Fig. 3) and incubated at 298 K for 5 min. The protocol of holo-TtuA and TtuB preparation was published previously[10]. The final concentration of TtuA was ~0.5 mM for the subsequent EPR spectroscopy measurements. Sample mixtures were incubated at 298 K for 10 min with a fivefold molar excess of sodium DT to reduce the Fe-S cluster, then transferred to a quartz EPR tube (Sigma, St. Louis, Missouri, USA) and frozen by immersion in liquid nitrogen in anaerobic conditions. Samples were stored in liquid nitrogen and shipped to the Analytical Research Center for Experimental Sciences, Saga University. CW X-band EPR spectra were obtained using a Bruker ELEXSYS E580 spectrometer, in continuous-wave mode operating at ~9.59 GHz, equipped with an Oxford Instruments ESR 910 continuous helium-flow cryostat. Experimental parameters were at 12 K, 1 mW microwave power, 100 kHz field modulation, and 10 G modulation amplitude. EPR simulation was generated by the program EasySpin ver. 5.2.20 (ref. [35]) operated in Matlab.

**Crystallization, data collection, and structure determination of holo-TtuA–TtuB in complex with ATP.** The holo-TtuA–TtuB solution was concentrated to 13.4 mg/ml for crystallization, which was conducted using the sitting drop vapor diffusion method at 293 K in anaerobic conditions. Crystals used for further experiments were grown in crystallization buffer (0.15 M ammonium sulfate, 0.1 M Tris at pH 8.0, and 15% w/v PEG 4000). The protein solution was mixed with crystallization buffer in volume of 0.5 μl to 0.5 μl and put on a well containing 75 μl of reservoir solution. The crystals were soaked in crystallization buffer containing 2.5 mM ATP and 20% v/v glycerol for 1 min, followed by soaking in crystallization buffer containing 5 mM ATP and 20% v/v glycerol for another 1 min. Subsequently, the crystals were frozen in liquid nitrogen. To maintain anaerobic conditions throughout data collection, the crystals were frozen in the anaerobic chamber and stored in liquid nitrogen until X-ray diffraction analysis.

X-ray diffraction data were collected at beamline BL41XU of SPring-8 (Harima, Japan) under cryogenic conditions (100 K) using X-rays of 1.0000 Å wavelength. Data were indexed, integrated, and scaled using the program XDS (ver. Jun 17, 2015)[36]. Data collection statistics are summarized in Table 1. The initial phase was

### Table 1 Data collection and refinement statistics.

| | holo-*Tth*TtuA-*Tth*TtuB-ATP complex |
|---|---|
| Data collection | |
| Space group | *C2* |
| Cell dimensions | |
| a, b, c (Å) | 165.6 82.8 138.5 |
| α, β, γ (°) | 90 116 90 |
| Resolution (Å) | 45.90–2.20 (2.28–2.20)[a] |
| [b]$R_{sym}$ | 4.6 (51.5) |
| $I/\sigma(I)$ | 15.01 (1.93) |
| Completeness (%) | 98.45 (96.00) |
| Redundancy | 3.46 (3.36) |
| Refinement | |
| Resolution (Å) | 45.9–2.20 (2.22–2.20) |
| No. reflections | 84554 |
| [c]$R_{work}$/[d]$R_{free}$ | 23.1 (31.3)/25.3 (33.4) |
| No. atoms | |
| Protein | 8894 |
| Ligand/ion | 164 |
| Water | 111 |
| B-factors | |
| Protein | 25.62 |
| Ligand/ion | 66.02 |
| Water | 45.78 |
| R.m.s. deviations | |
| Bond lengths (Å) | 0.003 |
| Bond angles (°) | 0.819 |

[a]Values in parentheses are for highest-resolution shell.
[b]$R_{mean} = \Sigma_{hkl}\Sigma_i|I_i(hkl) - \langle I(hkl)\rangle|/\Sigma_{hkl}\Sigma_i I_i(hkl)$, where $I_i(hkl)$ is the mean intensities of a set of equivalent reflections.
[c]$R_{work}(\%) = \Sigma_{hkl}||F_{obs}(hkl)| - |F_{calc}(hkl)||/\Sigma_{hkl}|F_{obs}(hkl)|$.
[d]$R_{free}(\%) = R_{work}$ calculated using 5% of the reflection data chosen randomly and omitted from the refinement.

determined by the molecular replacement method in the program Phaser MR (ver.2.5.6)[37] using the crystal structure of a TtuA–TtuB (G65C) mutant (PDB ID 5GHA) as a search model. Structure refinement was conducted using the program phenix.refine (ver.1.9)[38]. After sufficient cycles of refinement and manual model building using the program COOT (ver.0.7.2)[39,40], the crystallographic factors $R_{work}$ and $R_{free}$ converged to 23.1% and 25.3%, respectively. The statistical score of the Ramachandran plot shows that 99.0% are in the most favored regions, 1.0% in the allowed regions, and 1.0% in outliers regions. Other details of refinement statistics are summarized in Table 1.

**Evaluation of s²T formation by TtuA**. Evaluation of s²T formation on the tRNA substrate was conducted as described previously[10]. Briefly, 150 pmol holo-TtuA (or its mutant) and 600 pmol brewer's yeast total tRNA (Sigma, St. Louis, Missouri, USA) were incubated with a sulfur donor (300 nmol sodium sulfide, or 600 pmol TtuB-COSH) at 333 K in reaction buffer composed of 50 mM HEPES-KOH (pH 7.6), 100 mM KCl, 10 mM magnesium chloride, 0.1 mM dithiothreitol, and 2.5 mM ATP in anaerobic conditions. The total volume of the reaction mixture was 30 μl. After incubation for the determined time interval, the reaction was stopped by addition of 75 μl Isogen (Nippon Gene, Tokyo, Japan) and 45 μl deionized water. Subsequently, tRNA was extracted with phenol/chloroform (5/1, pH 4.5; Thermo Fisher, Waltham, Massachusetts, USA), precipitated with ethanol, digested with Nuclease P1 (Yamasa Co., Choshi, Japan), and alkaline phosphatase from bacteria (Takara Bio Inc., Kusatsu, Japan), then analyzed using an Inertsil ODS-3 column (2.1 × 150 mm × 3 μm; GL Science, Tokyo, Japan) connected to an Extrema HPLC system (Jasco, Tokyo, Japan). The amount of s²T was quantified using the peak area of pseudouridine (Ψ) as a reference (Supplementary Fig. 5).

**Confirmation of desulfurization from TtuB-COSH**. Desulfurization was confirmed by SDS–PAGE with gels containing 11.7 μg/ml of APM (Toronto Research Chemicals, North York, Ontario, Canada). An aliquot of 450–600 pmol of TtuB-COSH incubated in each specified condition was loaded onto each lane (Figs. 3–5). The extent of desulfurization was determined by the shift in mobility.

**Reporting summary**. Further information on research design is available in the Nature Research Reporting Summary linked to this article.

## Data availability

The atomic coordinates and structure factors of holo-TtuA in complex with TtuB and ATP have been deposited in the Protein Data Bank with accession number 5ZTB. The source data underlying Fig. 5b, c are shown in Supplementary Data 1 and 2. Full gels are shown in Supplementary Information. Plasmids are available upon request from the corresponding author, M.Y.

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

## Acknowledgements
X-ray diffraction experiments were conducted at the Photon Factory (proposals 2016G092 and 2015G067) and SPring-8 (proposals 2015B2114, 2015B2117, 2015B6524, and 2016B2565). We thank the beamline staff of SPring-8 and Photon Factory for their assistance. This work was supported by the Platform Project for Supporting Drug Discovery and Life Science Research (Basis for Supporting Innovative Drug Discovery and Life Science Research; BINDS) from the Japan Agency for Medical Research and Development grant number JP18am0101071 (to M.Y.), Hokkaido University Special Educational Program "Nitobe Scool" PKF8618101 (to M.I.), JST PRESTO JPMJPR1517 (to Y.T.), and Grants-in-Aid for Scientific Research (15J01961, to M.C.; 24000011, 19H02519, 19H03511, 19H03040, 19H02831, 19H00918, Bilateral Programs, and 19H02519, to Y.T.; 17H05424, to M.Y.; and 17H06955 and 18H02412 to M.H.) from the Ministry of Education, Culture, Sports, Science, and Technology of Japan; and NEDO (to Y.T.).

## Author contributions
M.C. performed the majority of experiments and drafted the manuscript. Y.T. supervised the work and wrote the manuscript. S.N. contributed to the structural studies and the desulfurization assay. M.I. contributed to the mutation study. M.H. performed EPR measurements and analyzed the data. N.S. performed the preliminary experiment of the mutation study and contributed to the research design. M.Y. supervised the work. All authors read and approved the final manuscript.

## Competing interests
The authors declare no competing interests.
