## [Peer Review File · Communications Biology]

Reviewers' comments:

Reviewer #1 (Remarks to the Author):

The described work is a solid contribution that advances knowledge of a very interesting system. The authors, however, make sweeping claims that are not justified by the scope of the work, and they fail to acknowledge reasonable alternatives to the proposed mechanism. Until those problems are eliminated in a revision, the manuscript does not merit publication.

Mechanistic Alternative. The authors are fully entitled to propose their favored mechanism, but they must acknowledge that reference 11 (nor any other of which I am aware) provides direct observation of the attractive but not yet verified [Fe₄S₅] intermediate. In an alternative that seems to fit their data equally well, the oxygen atom of the thiocarboxylate coordinates to the iron-sulfur cluster, which serves as an elaborate Lewis acid to position the thiocarboxylate as a nucleophile (to attack the adenylated tRNA) and also activate the carbonyl group for a subsequent nucleophilic attack by the water or hydroxide ion (see below) that is also coordinated to the cluster. Until a 'fully loaded' crystal structure that contains all reaction components and/or products, there is simply too much 'play' in the positioning of the components that are there to make sweeping conclusions and ignore plausible alternatives. The revision should include many places throughout the text, but the entire last paragraph needs to be rewritten from scratch.

Mechanistic Conclusions. The authors have most definitely not "revealed" the "molecular mechanism for sulfur transfer". Nor has "previous research . . . revealed how TtuA transfers the sulfur located on the tip of the Fe-S cluster to tRNA . . ." [That work did reveal electron density that can be modeled as either hydroxide or hydrosulfide—the authors cannot be allowed to then claim more in passing than the original paper demonstrates.] The manuscript lays out beautiful work that significantly advances understanding of the system, but the authors' claims must stay within the bounds of what is actually shown.

Role of Residue 137. The repeated claim that Lys137 is "responsible" for "extracting" sulfur needs to be restated. Clearly, residue 137 can be either a Lys or Arg, and the side chain groups of either allows the desulfurization reaction to proceed. However, common usage is not that the residue is "responsible", and the statement that the cluster "releases sulfur from Lys137" is absurd on its face (although likely due to a misunderstanding of language rather than chemistry). At least one sentence must be altered because Arg has a guanidino group rather than an amino group (as is stated). It seems quite likely to me that a coordinated hydroxide ion benefiting from the charge on the side chain of either Lys or Arg better explains the observations, and I do not think sufficient spectroscopy has been performed to distinguish a bound hydroxide ion versus a bound water molecule. The discussion of the ability of the variants to utilize bisulfide (rather than the whole system) is awkwardly phrased throughout as a restoration of activity rather than an ability to use an alternative sulfur source. The observations are informative and should definitely be told, but the telling needs to be more precise to achieve clarity.

Structure. I would prefer a single larger structure figure. I find it somewhat disappointing that the data that is most directly interpretable and easiest 'to see' component of this fine work is relegated to two small figure panels. I would also like to see (but cannot demand) a very brief discussion of the slight differences in O-Fe distances between the carboxylate oxygens. Is 2.3 versus 2.5 Å significant? A better view of the structure would also make it easier for the reader to judge whether one oxygen is truly coordinated and the other near the iron simply because of the bonding inherent within a carboxylate group (the oxygens MUST be within a set distance of each other): does the 2.5 Å spacing between Fe and O simply result from the distance constraints of the carboxylate group? That

discussion would fit into a fuller consideration of the alternative mechanism (see above), although the authors should also note how facile it would be to shift to a clearly monodentate coordination by the (thio)carboxylate in response to binding of the other reaction components or simple thermal fluctuations. Also, is the structural data sufficient to distinguish between two equally populated states that differ by which oxygen atom is in contact with the iron? I suggest that the authors reconfigure their figures to remove explicit double bonds among oxygens that are identical (due to resonance).

Lack of Clarity. Less concerning are several instances of imprecision in language that may confuse or give the wrong impression to a reader. p 7: The interpretation of the change in EPR signal intensity in the presence of the TtuB thiocarboxylate needs to be fleshed out. Why does the change in redox potential lead to the change in intensity—is it oxidation or reduction of the cluster that the authors are proposing? p. 8 What is "adequate" enzyme activity? p. 9 "Since these residues are located far away from the Fe-S and TtuB (Fig 4a), they are not directly involved in the desulfurization reaction, but probably have an indirect influence." If the authors are making some point beyond the obvious that distant residues cannot have a direct effect (which need not be stated), they must rewrite the sentence to make their point. p 10 "The intermediate formation of [4Fe-5S] . . . is considered to be essential for thiolation . . ." By whom? As stated above, if there is direct evidence for that attractive intermediate, state it succinctly (with a reference); otherwise, the authors must dial back the strength of their assertion that the intermediate is on one plausible pathway that is favored by them. Figure 4: A direct interaction is shown between a phosphate oxygen and Asp59. Is there any evidence that one or the other bears a proton? I also cannot judge the disposition of the two groups in three dimensions, but it seems likely that this case is yet another where a 'dumb' program uses only the inter-atomic distance rather than orientation to deduce a hydrogen bond (sometimes even when the hydrogen would clearly point in the wrong direction). Finally, I do not care for the use of "trigger" and "pause" to describe the lack of reaction merely upon coordination of the thiocarboxylate. Many enzymes gather all substrates before achieving a productive conformation, and many others prevent the attainment of a reactive constellation of active site groups until the reaction can 'safely' proceed (e.g., closure of a loop to insure that an intermediate cannot diffuse from the active site). The language—especially "pause"—simply does not strike me as apt but as an implicit claim of something more extraordinary than the phenomenon seems to be.

In closing, I reiterate my tremendous enthusiasm for both the work and its impact in shaping the understanding of these intriguing sulfur transfer reactions. I hope that authors take this constructive criticism in the spirit in which it is offered: to publish excellent work in a clearer manuscript that does not oversell the conclusions, which can lead to both dismissal by experts as overblown and a false sense by non-experts that more is established than is the case.

Reviewer #2 (Remarks to the Author):

The manuscript "Desulfurization of ubiquitin-like sulfur donor TtuB by the [4Fe-4S] cluster of sulfurtransferase TtuA" by Minghao Chen et al. provides molecular insights on TtuA-TtuB mechanism for the biosynthesis of s2T. Indeed, recent results in the literature showed that the [4Fe-4S] cluster of TtuA binds sulfide suggesting that thiolation of tRNA occurs via sulfur binding to the cluster of TtuA and transfer to the tRNA substrate and that sulfur transfer exists from TtuB-COSH to TtuA. However, there are no molecular data in favor of that mechanism. Using spectroscopy, x-ray crystallography and biochemical assays the authors demonstrate that the Fe-S cluster of TtuA directly receive sulfur from TtuB through its coordination ability. In addition, it was nicely demonstrated that within the TtuB-[4Fe-4S]TtuA intermediate, the release of sulfur from TtuB-COSH is dependent on adenylation of

the substrate. This paper provides important mechanistic data that will be usefull in the field of tRNA modification and in general of Fe-S clusters. The present manuscript deserves publication, without revision, to Communications Biology.

Response to referee 1

[Comment]

The described work is a solid contribution that advances knowledge of a very interesting system. The authors, however, make sweeping claims that are not justified by the scope of the work, and they fail to acknowledge reasonable alternatives to the proposed mechanism. Until those problems are eliminated in a revision, the manuscript does not merit publication.

[Response]

Thank you for your attentive comments from an expert perspective of sulfur transfer reaction. The main concern is regarding the mechanistic alternative, which was already discussed in our previous publication (Chen *et al.*, PNAS 2017). We have proposed two hypothetical functions of Fe-S cluster, i.e., 1) Fe-S extracts sulfur from TtuB and transfers it to substrate tRNA via forming a [4Fe-5S] intermediate, and 2) Fe-S stabilizes the C-terminus of TtuB and facilitate the direct transfer from TtuB to tRNA, which were named “indirect” and “direct” mechanism, respectively. The manuscript was written based on the former hypothesis for the reasons described below, and the “alternative mechanism” mentioned by referee 1 corresponds to the latter mechanism.

Considering our new readers, a comprehensive introduction including both hypothetical mechanisms is needed. In the revised manuscript, we added an introduction for this argument and reviewed both of them based on the new knowledge acquired in the present study.

Our point-by-point responses are as follows.

[Comment]

Mechanistic Alternative. The authors are fully entitled to propose their favored mechanism, but they must acknowledge that reference 11 (nor any other of which I am aware) provides direct observation of the attractive but not yet verified [Fe4S5] intermediate. In an alternative that seems to fit their data equally well, the oxygen atom of the thiocarboxylate coordinates to the iron-sulfur cluster, which serves as an elaborate Lewis acid to position the thiocarboxylate as a nucleophile (to attack the adenylated tRNA) and also activate the carbonyl group for a subsequent nucleophilic attack by the water or hydroxide ion (see below) that is also coordinated to the cluster. Until a 'fully loaded' crystal structure that contains all reaction components and/or products, there is simply too much 'play' in the positioning of the components that are there to make sweeping conclusions and ignore plausible alternatives. The revision should include many places throughout the text, but the entire last paragraph needs to be rewritten from scratch.

[Response]

As suggested by the reviewer, in the reference 11, an [4Fe-5S] intermediate was not verified but “proposed”. Therefore, we revised the description in the introduction section as follows.

Introduction, line 58-64

In a recent study using X-ray crystallography, an extra electron density was discovered bound to the unique Fe site of the [4Fe-4S] cluster of *Pyrococcus horikoshii* TtuA. The atom was not

identified but, because the electron density most closely resembled a hydrosulfide ion, Arragain et al. proposed that the unique Fe site of TtuA is responsible for capturing the sulfide ion and transferring it to the substrate tRNA¹¹. In their proposed mechanism, sulfur is first trapped on the unique Fe site of the [4Fe-4S] cluster (hereafter this state is referred to as the [4Fe-5S]-intermediate), from which it is transferred to tRNA.

According to the reviewer's comments, discussion was revised as follows. In particular, last paragraph was revised thoroughly.

Discussion, line 220-233

Mechanism of sulfur transfer from TtuB to tRNA via Fe-S cluster

The biosynthesis of s²T is composed of two sequential reactions: adenylation (Fig. 6(e) to (f)) and thiolation (Fig. 6 (f) to (g)). Adenylation of the substrate is commonly used by a large variety of enzymes. In general, the PP-loop domain is responsible for the adenylation and the Fe-S cluster is not required. Therefore, adenylation of the substrate tRNA by TtuA should be similar to that of other enzymes in which the Fe-S cluster does not contribute. For the later thiolation step, there are two probable pathways: (1) sulfur is at first trapped on the Fe-S cluster and then transferred to tRNA (hereafter referred to as the indirect pathway); and (2) sulfur is directly transferred from the thiocarboxylated C-terminus of TtuB to tRNA (hereafter referred to as the direct pathway). The indirect pathway (Fig. 6 (a) to (c)) is considered to be more reasonable because: (i) the sulfur donor protein TtuB is absent in some organisms, including *Thermotoga maritima* and *Pyrococcus horikoshii*; (ii) even TtuA that has a native-partner TtuB (e.g., *Thermus thermophilus* TtuA) can transfer a sulfide ion to tRNA; and (iii) an extra electron density (to which a sulfide ion fits well) was found on the unique Fe site of the [4Fe-4S] cluster of *P. horikoshii* TtuA¹¹.

Discussion, line 276-278

It should be noted that this mechanism assumes the indirect pathway for the sulfur transfer from TtuB-COSH to tRNA because it is the most plausible pathway considering previous experimental data.

Discussion, line 288-301

It is worth mentioning an alternative mechanism (the direct mechanism; Supplementary Information Fig. S6) whereby sulfur is considered to be transferred from a [4Fe-4S]-TtuB intermediate directly to tRNA. The free Fe site of the [4Fe-4S] cluster would be used to monodentately coordinate the oxygen atom of the thiocarboxylate group and proffer the sulfur to the substrate tRNA. The carbonyl carbon will be more positively charged due to coordination to the [4Fe-4S] cluster by the inductive effect, which may further enhance the release of sulfur to attack the adenylated tRNA. The subsequent elimination of sulfur from the carbonyl group would be carried out by a nucleophilic attack of the water coordinated between free Fe and Lys137. Based on the data obtained in the present study, the possibility that the direct mechanism occurs cannot be excluded. Indeed, it provides a more reasonable explanation than the indirect mechanism because the TtuB desulfurization reaction halts until the tRNA adenylation reaction takes place (Supplementary Information Fig.6 (b') to (f)). However, because the native-partner TtuB (crucial for the direct mechanism) does not exist in all the organisms in which this reaction occurs, it is probably not the primary mechanism.

[Comment]

Mechanistic Conclusions. The authors have most definitely not "revealed" the "molecular mechanism for sulfur transfer". Nor has "previous research . . . revealed how TtuA transfers

the sulfur located on the tip of the Fe-S cluster to tRNA" [That work did reveal electron density that can be modeled as either hydroxide or hydrosulfide—the authors cannot be allowed to then claim more in passing than the original paper demonstrates.] The manuscript lays out beautiful work that significantly advances understanding of the system, but the authors' claims must stay within the bounds of what is actually shown.

[Responses]

We reconsidered the phrase of “revealed the molecular mechanism for sulfur transfer” and thoroughly complemented the description of mechanistic alternative in the revised manuscript, as follows.

line 303-304

In conclusion, the present study provides insights into the involvement of the Fe-S cluster on desulfurization of TtuB-COSH.

line 309-314

The plausible reaction assuming the indirect pathway is summarized in Fig. 6. Previous research has found that either hydrosulfide or hydroxide was coordinated on the unique Fe site, which agrees well with the indirect pathway¹¹. For adenylation of the substrate tRNA, a general mechanism of ATP pyrophosphatase is considered to be used in TtuA^{32,33}. Together with these previous studies, the present study proposes the entire most-plausible reaction mechanism of desulfurization by TtuA from TtuB-COSH:

[Comment]

Role of Residue 137. The repeated claim that Lys137 is "responsible" for "extracting" sulfur needs to be restated. Clearly, residue 137 can be either a Lys or Arg, and the side chain groups of either allows the desulfurization reaction to proceed. However, common usage is not that the residue is "responsible", and the statement that the cluster "releases sulfur from Lys137" is absurd on its face (although likely due to a misunderstanding of language rather than chemistry). At least one sentence must be altered because Arg has a guanidino group rather than an amino group (as is stated). It seems quite likely to me that a coordinated hydroxide ion benefiting from the charge on the side chain of either Lys or Arg better explains the observations, and I do not think sufficient spectroscopy has been performed to distinguish a bound hydroxide ion versus a bound water molecule. The discussion of the ability of the variants to utilize bisulfide (rather than the whole system) is awkwardly phrased throughout as a restoration of activity rather than an ability to use an alternative sulfur source. The observations are informative and should definitely be told, but the telling needs to be more precise to achieve clarity.

[Responses]

According to the reviewer's comments, we rephrased the descriptions as follows.

line 260-261

it was deduced that the side chain of Lys137 is responsible for extracting sulfur from TtuB.

line 261-262

amino or guanidino group at the position of residue 137 would be necessary for extracting sulfur from TtuB-COSH.

line 315-316

(3) releases sulfur from TtuB by nucleophilic attack by a hydroxide ion, benefitting from a charge on the side chain at Lys137.

Regarding a spectroscopy, the spectroscopic study was performed in the purpose of detecting the change of coordination of Fe-S cluster in the presence of TtuB, not for distinguishing hydroxide/water molecule. For achieving the purpose, advanced techniques such as deuterium labeling and neutron diffraction are necessary.

Regarding the discussion of the activity of the variants, the descriptions were revised as follows. To make the discussion clearer, s^2T formation activity of K137R using Na_2S as a sulfur source was removed from Fig. 5c.

line 196-202

These results indicate that the side chain of the Lys137 is responsible for the extraction of sulfide from TtuB-COSH. That is, although the sulfur-extracting activity of the K137A mutant was diminished due to the absence of the side chain, s^2T formation activity was restored because the extraction of sulfur from TtuB-COSH is not necessary in the presence of Na_2S . It is noted that K137R showed enzymatic activity using TtuB-COSH as a sulfur donor (K137R of Fig. 5b). These results suggest that a positively charged side chain at position 137 is essential for the sulfur extracting activity of TtuA.

[Comment]

Structure. I would prefer a single larger structure figure. I find it somewhat disappointing that the data that is most directly interpretable and easiest 'to see' component of this fine work is relegated to two small figure panels.

I would also like to see (but cannot demand) a very brief discussion of the slight differences in O-Fe distances between the carboxylate oxygens. Is 2.3 versus 2.5 Å significant? A better view of the structure would also make it easier for the reader to judge whether one oxygen is truly coordinated and the other near the iron simply because of the bonding inherent within a carboxylate group (the oxygens MUST be within a set distance of each other): does the 2.5 Å spacing between Fe and O simply result from the distance constraints of the carboxylate group? That discussion would fit into a fuller consideration of the alternative mechanism (see above), although the authors should also note how facile it would be to shift to a clearly monodentate coordination by the (thio)carboxylate in response to binding of the other reaction components or simple thermal fluctuations.

Also, is the structural data sufficient to distinguish between two equally populated states that differ by which oxygen atom is in contact with the iron? I suggest that the authors reconfigure their figures to remove explicit double bonds among oxygens that are identical (due to resonance).

[Responses]

According to the reviewer's suggestion, Fig. 2 was newly prepared as an independent figure for showing structure. There contains entire structure of TtuA-TtuB-[Fe-S]-ATP complex and close-up view. Accordingly, the descriptions concerning structure were separated from the section of "Identification of [4Fe-4S]-TtuB intermediate" as a new section "Crystal structure of TtuA-TtuB-[Fe-S]-ATP complex" (line 101-168).

The distances of two carboxylate oxygens with Fe were revised to 2.4 and 2.5 Å based on the present coordination (PDBID: 5ZTB). The geometry was determined by refinement software

PHENIX.REFINE automatically. During refinement, we applied restraint for stereochemistry within each molecule, but not for distances and angles between molecules. For clarity, stereo structure is shown for close-up view (Fig. 2c). Although we, at present, cannot argue about the significance of the difference between 2.5 and 2.4 angstrom, it might be possible that the unique Fe site specifically bind with carbonyl oxygen and proffer the sulfur to the substrate tRNA. This binding manner can also facilitate the release of the sulfur atom from TtuB in a perspective of organic chemistry. As the reviewer suggests, our crystal structure cannot exclude the possibility of this alternative mechanism. Therefore, the following descriptions were added.

line 113-114

The distance between each of the two oxygen atoms and the unique Fe site was 2.4 and 2.5 Å, respectively (Fig. 2c).

line 290-296

The free Fe site of the [4Fe-4S] cluster would be used to monodentately coordinate the oxygen atom of the thiocarboxylate group and proffer the sulfur to the substrate tRNA. The carbonyl carbon will be more positively charged due to coordination to the [4Fe-4S] cluster by the inductive effect, which may further enhance the release of sulfur to attack the adenylated tRNA. The subsequent elimination of sulfur from the carbonyl group would be carried out by a nucleophilic attack of the water coordinated between free Fe and Lys137.

We cannot distinguish difference between double bond and single bond. The double bonds were due to the default setting of graphic software. As suggested by the reviewer, explicit double bonds among oxygens were removed.

[Comment]

Lack of Clarity. Less concerning are several instances of imprecision in language that may confuse or give the wrong impression to a reader. p 7: The interpretation of the change in EPR signal intensity in the presence of the TtuB thiocarboxylate needs to be fleshed out. Why does the change in redox potential lead to the change in intensity—is it oxidation or reduction of the cluster that the authors are proposing?

[Responses]

According to reviewer's comments, we added our interpretation as below.

Line 160-163

This would be caused by alteration of the redox potential of the Fe-S cluster because the cluster became more likely to be reduced by dithionite and the quantity of reduced [4Fe-4S] cluster was increased when TtuB-COSH was bound

[Comment]

p. 8 What is "adequate" enzyme activity?

[Responses]

"Adequate" means the enzyme activity is obvious which can't be treated as an experimental error. We delated this word for making the phrase more straightforward.

[Comment]

p. 9 "Since these residues are located far away from the Fe-S and TtuB (Fig 4a), they are not directly involved in the desulfurization reaction, but probably have an indirect influence." If the authors are making some point beyond the obvious that distant residues cannot have a direct effect (which need not be stated), they must rewrite the sentence to make their point.

[Responses]

We have revised this paragraph as follows.

line 210-217

The ability of S55A, D59A, and D161A to remove sulfur from TtuB-COSH was further evaluated and found to be significantly lower for all three mutants (Fig. 5d). In the crystal structure, Ser55, Asp59, and Asp161 are located out of range of direct interaction with the carboxyl terminus of TtuB (farther than 7 Å), suggesting that diminished desulfurization of these mutants was not due to an inability for direct interaction with the C-terminus of TtuB. These residues are known to be involved in ATP hydrolysis, as mentioned above³³. It is therefore plausible that the abolition of ATP hydrolytic activity (owing to substitution for Ser55, Asp59, and Asp161) caused the dramatic observed decrease of desulfurization.

[Comment]

p 10 "The intermediate formation of [4Fe-5S] . . . is considered to be essential for thiolation . . ." By whom? As stated above, if there is direct evidence for that attractive intermediate, state it succinctly (with a reference); otherwise, the authors must dial back the strength of their assertion that the intermediate is on one plausible pathway that is favored by them.

[Responses]

The [4Fe-5S] intermediate based reaction mechanism was proposed by Arragain *et al.* (Reference 11). Although there is yet no direct evidence to prove the intermediate, based on the following facts (i) sulfur donor protein TtuB is absent in some organisms including *Thermotoga maritima* and *Pyrococcus horikoshii*, (ii) even TtuA which has native partner TtuB (e.g., *Thermus thermophilus* TtuA) can transfer sulfide ion to tRNA, and (iii) an extra electron density to which sulfide ion fit well was found on the unique Fe site of the [4Fe-4S] cluster of *P. horikoshii* TtuA, the [4Fe-5S] intermediate mechanism is the most reasonable candidate at the moment. The description on p 10 was revised as follows.

line 224-237

Mechanism of sulfur transfer from TtuB to tRNA via Fe-S cluster

The biosynthesis of s²T is composed of two sequential reactions: adenylation (Fig. 6(e) to (f)) and thiolation (Fig. 6 (f) to (g)). Adenylation of the substrate is commonly used by a large variety of enzymes. In general, the PP-loop domain is responsible for the adenylation and the Fe-S cluster is not required. Therefore, adenylation of the substrate tRNA by TtuA should be similar to that of other enzymes in which the Fe-S cluster does not contribute. For the later thiolation step, there are two probable pathways: (1) sulfur is at first trapped on the Fe-S cluster and then transferred to tRNA (hereafter referred to as the indirect pathway); and (2) sulfur is directly transferred from the thiocarboxylated C-terminus of TtuB to tRNA (hereafter referred to as the direct pathway). The indirect pathway (Fig. 6 (a) to (c)) is considered to be more reasonable because: (i) the sulfur donor protein TtuB is absent in some organisms, including *Thermotoga maritima* and *Pyrococcus horikoshii*; (ii) even TtuA that has a native-partner TtuB (e.g., *Thermus thermophilus* TtuA) can transfer a sulfide ion to tRNA; and (iii) an extra electron

density (to which a sulfide ion fits well) was found on the unique Fe site of the [4Fe-4S] cluster of *P. horikoshii* TtuA¹¹ .

[Comment]

Figure 4: A direct interaction is shown between a phosphate oxygen and Asp59. Is there any evidence that one or the other bears a proton? I also cannot judge the disposition of the two groups in three dimensions, but it seems likely that this case is yet another where a 'dumb' program uses only the inter-atomic distance rather than orientation to deduce a hydrogen bond (sometimes even when the hydrogen would clearly point in the wrong direction).

[Responses]

Hydrogen bond formation between Asp59 and phosphate was confirmed by the program CONTACT. This program considers both distances and angles. In this case, the distance between the carbonyl oxygen of Asp59 and the phosphate oxygen is 2.7 Å, and the angles formed by carbonyl carbon - carbonyl oxygen – phosphate oxygen and carbonyl oxygen - phosphate oxygen – phosphorus are 108° and 120°, respectively, which correspond to the ideal hydrogen bond. Although electron density of hydrogen is not visible in this resolution, according to the appropriateness of the distance and angle, it is enough reasonable to conclude that there is a direct interaction between Asp59 and phosphate. Moreover, both Asp59 and Ser60 are belonging to the highly conserved PP-loop motif involved in ATP hydrolysis. Asp59 and Ser60 are responsible for direct binding to the gamma-phosphate of ATP in general (Fellner, Crit Rev Biochem Mol Biol 2018).

[Comment]

Finally, I do not care for the use of "trigger" and "pause" to describe the lack of reaction merely upon coordination of the thiocarboxylate. Many enzymes gather all substrates before achieving a productive conformation, and many others prevent the attainment of a reactive constellation of active site groups until the reaction can 'safely' proceed (e.g., closure of a loop to insure that an intermediate cannot diffuse from the active site). The language—especially "pause"—simply does not strike me as apt but as an implicit claim of something more extraordinary than the phenomenon seems to be.

[Responses]

According to the reviewer's suggestion, "trigger" and "pause" was rephrased by other words. In addition, these words were removed from Fig. 5.

Response to referee 2

[Comment]

The manuscript "Desulfurization of ubiquitin-like sulfur donor TtuB by the [4Fe-4S] cluster of sulfurtransferase TtuA" by Minghao Chen et al. provides molecular insights on TtuA-TtuB mechanism for the biosynthesis of s2T. Indeed, recent results in the literature showed that the [4Fe-4S] cluster of TtuA binds sulfide suggesting that thiolation of tRNA occurs via sulfur binding to the cluster of TtuA and transfer to the tRNA substrate and that sulfur transfer exists from TtuB-COSH to TtuA. However, there are no molecular data in favor of that mechanism. Using spectroscopy, x-ray crystallography and biochemical assays the authors demonstrate that the Fe-S cluster of TtuA directly receive sulfur from TtuB through its coordination ability. In addition, it was nicely demonstrated that within the TtuB-[4Fe-4S]TtuA intermediate, the release of sulfur from TtuB-COSH is dependent on adenylation of the substrate. This paper provides important mechanistic data that will be useful in the field of tRNA modification and in general of Fe-S clusters. The present manuscript deserves publication, without revision, to Communications Biology.

[Responses]

We appreciate your high evaluation of our work. We are looking forward to having the manuscript to be published in the near future and promote other studies on related fields including post-transcriptional modification, sulfur metabolism, and inorganic biochemistry.

REVIEWERS' COMMENTS:

Reviewer #2 (Remarks to the Author):

The largest deficiencies in this report of interesting findings on an intriguing system have been rectified by a discussion of an alternate mechanism and clarifications throughout the manuscript.

The new/revised figures greatly improve this submission compared to the original. Please add an explicit statement to the legend for Figure 5a that a proton (not shown) lies between Asp-59 and the gamma-phospho group of ATP. In Figure 6, add a negative charge to the three depictions of the TtuB C-terminus; as a chemist, I am unsettled by the abbreviation of AMP and suggest the authors at least include the oxygen attached to C2 if not Ado attached to an explicitly drawn phospho group.

Several points of imprecision remain and must be addressed. The most important are:

p 3 lines 60-61 "the electron density most closely resembled a hydrosulfide ion" should be something akin to "the electron density was more consistent with a hydrosulfide rather than a hydroxide ion"

p 8 line 169 "desulfurization state" is a non-starter; "sulfur content of TtuB" is more accurate.

throughout "extraction" to describe the transfer of sulfur is decidedly nonstandard; "mobilize" and "mobilization" are more typically used

p 9 lines 195-196 The authors retain the misleading usage of K137A "restoration[ing]" and being "responsible" for sulfur mobilization. The former is simply wrong: K137A TtuA is the only variant that is able to use bisulfide as an alternative source of sulfur to complete the overall reaction, and there is no 'restoration'. Lys137 is also NECESSARY (or "essential") for sulfur mobilization.

p 13 line 291 I have never seen "monodentate" converted to an adverb nor "proffer" used in a mechanistic context.

p 14 lines 313-317 The sentence construction here is mangled and requires a complete rewrite—I ask the copy editor(s) to assist the authors.

Response to referee 2

[Comment]

The largest deficiencies in this report of interesting findings on an intriguing system have been rectified by a discussion of an alternate mechanism and clarifications throughout the manuscript.

The new/revised figures greatly improve this submission compared to the original. Please add an explicit statement to the legend for Figure 5a that a proton (not shown) lies between Asp-59 and the gamma-phospho group of ATP. In Figure 6, add a negative charge to the three depictions of the TtuB C-terminus; as a chemist, I am unsettled by the abbreviation of AMP and suggest the authors at least include the oxygen attached to C2 if not Ado attached to an explicitly drawn phospho group.

[Response]

We are glad to have your high evaluation. The improvement of the revised manuscript would not be achieved without your attentive comments. We appreciate it very much.

Regarding your latest comments, the following changes are updated:

1. An explicit statement regarding the proton was added to the legend of Figure 5.
2. Signs of negative charge were added to the TtuB C-terminuses in Figure 6 and Supplementary Figure 4.
3. The abbreviation of AMP was replaced with adenosine (Ado) attached to a phosphoryl group in Figure 6 and Supplementary Figure 4.

.....
[Comment]

Several points of imprecision remain and must be addressed. The most important are:

p 3 lines 60-61 "the electron density most closely resembled a hydrosulfide ion" should be something akin to "the electron density was more consistent with a hydrosulfide rather than a hydroxide ion"

[Response]

Line 76-77, the phrase was revised to "the electron density was more consistent with a hydrosulfide rather than a hydroxide ion".

.....
[Comment]

p 8 line 169 "desulfurization state" is a non-starter; "sulfur content of TtuB" is more accurate. throughout "extraction" to describe the transfer of sulfur is decidedly nonstandard; "mobilize" and "mobilization" are more typically used

[Response]

Line 197, the phrase was revised to "sulfur content of TtuB".

The words “extract/extraction” describing TtuB desulfurization were replaced with “mobilize/mobilization” through the manuscript.

.....

[Comment]

p 9 lines 195-196 The authors retain the misleading usage of K137A "restoration[ing]" and being "responsible" for sulfur mobilization. The former is simply wrong: K137A TtuA is the only variant that is able to use bisulfide as an alternative source of sulfur to complete the overall reaction, and there is no 'restoration'. Lys137 is also NECESSARY (or "essential") for sulfur mobilization.

[Response]

Line 229, the phrase was revised to “K137A is the only variant that is able to use Na₂S as an alternative source of sulfur to complete the overall reaction among above four alanine-substituted mutants”

The word “responsible” for describing the role of K137 was replaced with “essential” through the manuscript.

.....

[Comment]

p 13 line 291 I have never seen "monodentate" converted to an adverb nor "proffer" used in a mechanistic context.

[Response]

Line 345, the word “monodentately” was deleted.

Line 346, the word “proffer” was replaced with “present”.

.....

[Comment]

p 14 lines 313-317 The sentence construction here is mangled and requires a complete rewrite—I ask the copy editor(s) to assist the authors.

[Response]

The manuscript had already been revised by a native speaker of English, who has a strong background in life science (a full Professor).

Response to editorial office

Please find the submitted checklist-table file for the point-to-point response.